# Genome-Wide Association Studies on the Kernel Row Number in a Multi-Parent Maize Population

**DOI:** 10.3390/ijms25063377

**Published:** 2024-03-16

**Authors:** Yizhu Wang, Fengyun Ran, Xingfu Yin, Fuyan Jiang, Yaqi Bi, Ranjan K. Shaw, Xingming Fan

**Affiliations:** 1College of Agronomy and Biotechnology, Yunnan Agricultural University, Kunming 650201, China; wang4710@outlook.com (Y.W.); ranfenfyun@outlook.com (F.R.); 2Institute of Food Crops, Yunnan Academy of Agricultural Sciences, Kunming 650205, China; xingfuyin626@163.com (X.Y.); jiangfuyansxx@126.com (F.J.); biyq122627@163.com (Y.B.); ranjanshaw@gmail.com (R.K.S.)

**Keywords:** maize, candidate gene, GWAS, QTL, KRN, GO/KEGG analysis, Mo17

## Abstract

Kernel row number (KRN) is a crucial trait in maize that directly influences yield; hence, understanding the mechanisms underlying KRN is vital for the development of high-yielding inbred lines and hybrids. We crossed four excellent panicle inbred lines (CML312, CML444, YML46, and YML32) with Ye107, and after eight generations of selfing, a multi-parent population was developed comprising four subpopulations, each consisting of 200 lines. KRN was accessed in five environments in Yunnan province over three years (2019, 2021, and 2022). The objectives of this study were to (1) identify quantitative trait loci and single nucleotide polymorphisms associated with KRN through linkage and genome-wide association analyses using high-quality genotypic data, (2) identify candidate genes regulating KRN by identifying co-localized QTLs and SNPs, and (3) explore the pathways involved in KRN formation and identify key candidate genes through Gene Ontology (GO) and Kyoto Encyclopedia of Genes and Genomes (KEGG) enrichment analyses. Our study successfully identified 277 significant Quantitative trait locus (QTLs) and 53 significant Single Nucleotide Polymorphism (SNPs) related to KRN. Based on gene expression, GO, and KEGG analyses, SNP-177304649, SNP-150393177, SNP-135283055, SNP-138554600, and SNP-120370778, which were highly likely to be associated with KRN, were identified. Seven novel candidate genes at this locus (*Zm00001d022420*, *Zm00001d022421*, *Zm00001d016202*, *Zm00001d050984*, *Zm00001d050985*, *Zm00001d016000*, and *Zm00014a012929*) are associated with KRN. Among these, *Zm00014a012929* was identified using the reference genome Mo17. The remaining six genes were identified using the reference genome B73. To our knowledge, this is the first report on the association of these genes with KRN in maize. These findings provide a theoretical foundation and valuable insights into the genetic mechanisms underlying maize KRN and the development of high-yielding hybrids through heterosis.

## 1. Introduction

Kernel row number (KRN) is a critical factor in maize yield and is an important trait for improving maize inbreds and developing high-yielding hybrids. The KRN of maize is a quantitative trait with high heritability that serves as a key indicator for evaluating the quality of inbred lines. Therefore, gathering genetic information about KRN is of great significance for maize breeding. Previous studies have shown that KRN is a quantitative trait controlled by multiple genes, and over 100 KRN-related QTLs have been identified in maize [1,2].

Since Thornsberry et al. [3] successfully introduced the genome-wide association study (GWAS) method into plant genetics research, GWAS has become a widely adopted method in various crops to identify loci associated with traits of interest. Maize, a cross-pollinated crop with rapid linkage disequilibrium decay, is well suited for association analysis. The development of molecular markers has significantly enhanced the effectiveness of QTL mapping for KRN [4]. Numerous studies have employed QTL mapping for KRN, revealing that the loci influencing KRN are distributed across all ten maize chromosomes. Using different mapping populations, more than 200 QTLs associated with KRN have been identified [5,6,7,8,9,10,11,12,13,14]. For instance, Lu et al. [13] conducted a study using a population of 397 F2:3 individuals across seven different environments and detected a significant KRN-related QTL, QTLqKRN7. Tian et al. [15] detected a major KRN-associated QTL, *qKRN10*, using three BC5F2:3 populations. Cai et al. [16] used an F2:3 population to identify three major QTLs associated with KRN in three different environments. In addition to identifying QTLs associated with KRN, researchers have conducted fine mapping to narrow the QTL interval, leading to the discovery of candidate genes. Calderon et al. [17] used a BC2S3 population to locate a KRN-related QTL and identified seven candidate genes located within a 203 kb region in the QTL. Bai et al. [18] detected a KRN-related QTL, qKRN5.04, and 10 candidate genes within a 300 kb region of the QTL. Although several candidate genes associated with KRN have been identified, only a limited number of functional genes associated with this trait have been successfully cloned. Liu et al. [19] and Wang et al. [20] achieved significant milestones by fine mapping and cloning *KRN4* and *KRN1*, respectively, in maize. Furthermore, Chen et al. [21] made a notable contribution by identifying *KRN2*, a gene that displays variation between domesticated maize and its wild ancestor, teosinte. *KRN2* serves as a pivotal quantitative trait locus for kernel row number in maize, with selection pressures in noncoding upstream regions resulting in decreased *KRN2* expression. This reduction correlates with an augmented grain number, which is facilitated by an increase in the KRN.

Previous studies have revealed that relying solely on a single approach, such as GWAS or QTL mapping, has significant limitations for identifying candidate genes that regulate KRN. In recent years, researchers have increasingly combined two or more mapping methods to mine candidate genes associated with target traits. Xiao et al. [22] mapped the QTLs affecting ear traits in ten different recombinant inbred line (RIL) populations using three methods. They then validated and fine-mapped four new QTLs using candidate gene association analysis, expression QTL analysis, and heterozygous inbred line validation. They successfully fine-mapped the QTLs associated with KRN. Similarly, Liu et al. [23] employed a combination of linkage and association mapping to unravel the genetic architecture of maize KRN and evaluated phenotypic predictability using the detected loci. A GWAS study revealed 31 associated SNPs representing 17 genomic loci with effects in at least one of the five individual environments. In another study, Fei et al. [24] utilized a nested association mapping (NAM) population consisting of 1617 RILs and identified five QTLs related to KRN via joint linkage mapping (JLM). These QTLs were further validated through linkage mapping (SLM) and GWAS. They identified three cloned KRN genes through a QTL assay, and two new KRN-related QTLs, qKRN4.2 and qKRN9.1, were successfully identified. Understanding the genetic basis of maize KRN and identifying candidate genes that control KRN are crucial for breeding high-yielding inbred lines and hybrids. Although previous studies have identified numerous KRN-associated genes [25,26,27,28,29,30,31,32,33,34,35], few have applied GO and KEGG analyses in GWAS and QTL studies for candidate gene selection. Additionally, studies on the association between KRN and tropical inbred lines are limited. In this study, four excellent inbred lines with high genetic variation in KRN (CML312, CML444, YML32, and YML46) were selected from the Suwan and non-Reid heterotic groups of tropical regions as donor parents. These inbred lines were hybridized with the elite maize inbred line, Ye107, from the Reid heterotic group, to develop a multi-parent population. The KRN phenotypic data were collected in five different environments using this multi-parent population. Subsequently, GWAS analysis and QTL mapping were conducted on the 655 RILs, followed by GO and KEGG enrichment analyses. The objectives of this study were to (1) utilize high-quality genotypic data to perform linkage and GWAS analyses of KRN in a multi-parent maize population to identify QTLs and SNPs associated with maize KRN, (2) identify candidate genes associated with KRN based on co-localized SNP and QTL information, and (3) combine the results of GO and KEGG enrichment analyses to identify the pathways involved in KRN formation and identify key candidate genes.

## 2. Results

### 2.1. Phenotypic Analysis of KRN in the Multi-Parent Population

KRN data were collected from a multi-parent population consisting of 655 RILs across five different environments. The environments were as follows: two in 2019 at Dehong (19DH) and Baoshan (19BS), two in 2021 at Jinghong (21JI) and Yanshan (21YS), and one in 2022 at Yanshan (22YS). As shown in Table 1, the range of KRN variation in 19DH was from 6.5 to 16.67, with an average KRN of 12.16. For 21YS, the range was from 6 to 16, with an average of 11.9; for 22YS, the range was from 9 to 19.5, with an average of 12.76. At 21JH, the range spanned from 8 to 20, with an average of 13.24. In 19BS, the range was 5.33 to 16.67 with an average of 11.82. Phenotypic data across the environments revealed high broad-sense heritability for KRN, reaching 61.94% (Table 1, Figure 1).

### 2.2. Structure Analysis of the Multi-Parent Population

To mitigate the impact of population stratification during GWAS analysis, we employed a mixed linear model (MLM) that incorporated both the population structure (Q) and kinship matrix (K) to reduce false positives, as suggested in a previous study [36]. We analyzed 590,816 high-quality SNPs using the Plink v1.9 software and performed a genetic similarity analysis using Tassel5. The resulting phylogenetic tree grouped the 655 RILs into four subpopulations (Figure 2). Additionally, we conducted principal component analysis (PCA) using R v 4.3.0 software, which initially divided the 655 RILs into four populations, labeled sub-pop1 to sub-pop4 (Figure 2). However, because the sub-pop4 and sub-pop1 groups overlapped, they were merged into one population. Thus, the 655 RILs were classified into three groups: sub-pop2, sub-pop3, and sub-pop1_sub-pop4 (Figure 3). This result is consistent with the “three heterotic group” theory in maize that has been used to improve breeding efficiency [37,38,39]. This also confirmed the accuracy of classifying the maize lines into different heterotic groups based on their combining ability and grain yield.

### 2.3. Genome-Wide Association Analysis for KRN

We obtained genotyping data for 590,816 high-quality SNPs for GWAS analysis to identify the loci associated with KRN. GWAS was performed using both genotyping and phenotypic data from the five environments, utilizing the MLM model implemented in GEMMA software (https://github.com/genetics-statistics/GEMMA/releases, accessed on 1 June 2023). Using the formula −log10 (1/SNP number), we established a significant threshold of five. By applying this threshold, we identified 53 SNPs associated with KRNs across five environments associated with 38 genes. (Figure 4a–d and Appendix A). Based on our stringent significant threshold, we identified the SNPs across various environments. In the 19DH environment, six SNPs were detected that were located on chromosomes 5 and 7. These SNPs collectively accounted for 8.8% and 6.4% of the phenotypic variance explained (PVE), respectively. Additionally, one SNP located on chromosome 6 in the 19BS environment contributed to 3.2% of the PVE. Furthermore, two SNPs were identified in the 21YS environment, located on chromosomes 2 and 8. In the 21JH environment, 41 SNPs were identified that were distributed across chromosomes 1, 2, 3, 4, 5, 6, 8, and 10, explaining a phenotypic variance ranging from 4.7% to 21%. In the 22YS environment, three SNPs were identified on chromosome 10, explaining 14.9% of PVE. Notably, these SNPs were not co-localized across the environments.

### 2.4. Detection of Candidate Genes

In the GWAS analysis, 53 significant SNPs were identified, and these SNPs explained KRN variation ranging from 3.24% to 21.01% (Appendix A). Among these SNPs, seven had a phenotypic variation explained (PVE) of over 10%, and 14 genes were located within the range of 20 kb upstream and downstream of these seven SNPs (Table 2). In maize, KRN is largely influenced by the following tissues: meristem_16–19_day, ear primordium (2–4 mm), ear primordium (6–8 mm), and female spikelets. Among the genes listed in Table 3, *Zm00001d016202* encodes an E3 ubiquitin protein ligase found in ear primordium [40], meristem, and COP1 proteins. COP1 is crucial in plant biology and participates in light signal transduction and growth regulation as an E3 ubiquitin ligase. It interacts with various proteins to control the expression of light-responsive genes, thereby influencing photomorphogenesis, flowering time, and photoperiod-related development. Based on functional annotation and expression analysis, *Zm00001d016202* is regarded as a potential candidate gene regulating kernel row number (KRN). However, the remaining 13 genes did not appear to play a specific role in KRN regulation.

Through a search within 20 kb upstream and downstream of SNP-177304649, two genes, *Zm00001d022420* and *Zm00001d022421*, were found to be co-localized in the 19DH environment. Although *Zm00001d022420* was not expressed in the meristem_16–19_day or ear primordium, it showed an expression level of 61.5 in female spikelets. In contrast, *Zm00001d022421* was expressed in all four tissues, with the highest expression level among all genes. Expression levels were 122.7, 156, 124.5, and 464.8 FPKM in meristem_16–19, ear primordium (2–4 mm), ear primordium (6–8 mm), and female spikelets, respectively. Both *Zm00001d022420* and *Zm00001d022421* encode ricin B-like lectins, which are carbohydrate-binding agglutinins involved in binding to polysaccharides with Lewis X, Lewis Y, or lacto-N-aminoglycan structures through their agglutinin domains [53] Additionally, by comparing with previously published genes related to ear development, we identified a *bd1* (branched silkless1) gene located approximately 1 Mb downstream of SNP-177304649. The *bd1* gene leads to increased branching in the male ear, deformities, and lack of filaments in the female ear [54,55]. Therefore, it is speculated that *Zm00001d022420* and *Zm00001d022421* may contribute to the reduction in KRN in maize.

To confirm the two candidate genes, *Zm00001d022420* and *Zm00001d022421*, we conducted haplotype analysis using the Haploview 4.2 software (Figure 5). *Zm00001d022420* exhibited two haplotypes: HAP-1 (TAA) and HAP-2 (CGC). The results indicated that in 19DH environments, HAP-2 (CGC) was not significantly higher than HAP-1 (TAA), and *Zm00001d022420* did not have a predominant haplotype. *Zm00001d022421* contained two haplotypes: HAP-1 (AA) and HAP-2 (GG). In the 19DH environment, HAP-1 (AA) was significantly more prevalent than HAP-2 (GG) was. Therefore, HAP-1 (AA) was considered to be the dominant haplotype for *Zm00001d022421*.

### 2.5. Linkage Analysis

#### 2.5.1. Parental Line Variation for KRN

To facilitate the mapping of QTLs controlling KRN, this study selected five inbred lines with substantial differences in KRN as parents for constructing the multi-parent population. The KRN values of the parental lines were as follows: Ye107 had 8 rows, CML312 had 16 rows, CML444 had 16 rows, YML32 had 14 rows, and YML46 had 14 rows.

#### 2.5.2. Construction of Linkage Maps

We constructed genetic linkage maps for two subpopulations (sub-pop3 and sub-pop4) made from crosses between the common parent Ye107, which exhibited the lowest KRN, and CML312 and CML444, both of which had the highest KRNs of 16. RILs from the two subpopulations were filtered with a k-mer completeness of 0.8 and a deviation segregation of 0.001. The filter markers were then grouped into bins, each containing 15 markers without linkage. JoinMap software (v 4.0) was used to sort the bin markers for each subpopulation using the maximum likelihood method, and genetic distances were calculated using the Kosambi function. Ultimately, 1802 bin markers were assigned to the two maps (Appendix A). The linkage map was visualized using the Perl SVG module, which depicts the distribution of bins on each chromosome (Figure 6). The linkage map of sub-pop3 had a total length of 1045.83 cM with an average marker distance of 1.07 cM. The longest chromosome was chromosome 1, spanning 169 cM, and the shortest was chromosome 10, measuring 33.23 cM. The linkage map of sub-pop4 had a total length of 828.27 cM with an average marker spacing of 1.01 cM. The longest chromosome was chromosome 4 with a length of 151.08 cM, and the shortest was chromosome 9 with a length of 41.88 cM.

Collinearity analysis was performed by comparing marker positions on the genetic map (Appendix A) with a physical map of the two RIL populations. The order of most markers in both linkage maps in Appendix A is consistent with each chromosome, indicating strong collinearity and high mapping accuracy during linkage mapping in this study.

#### 2.5.3. QTL Mapping

QTL mapping was performed for KRN in two subpopulations (sub-pop3 and sub-pop4) of the multi-parent population across five different environments (19DH, 19BS, 21YS, 21JH, and 22YS). The best linear unbiased prediction (BLUP) for KRN was predicted for all environments. Subsequently, QTL mapping was conducted for BLUP and all individual environments for the two subpopulations using WinQTLcart 2.5 software and the composite interval mapping method (Figure 7). The LOD threshold was determined based on 1000 permutations with a significance level of *p* ≤ 0.05 [56]. QTLs with LOD scores ≥ 2.5 were considered significantly associated with KRN. Using this threshold, 178 QTLs were detected in five environments and BLUP for sub-pop3. These QTLs were distributed across chromosomes 1, 3, 4, 5, 7, and 9 (Appendix A). Marker 95 on chromosome 1 in 19DH had the highest LOD (6.299). In sub-pop3, chromosome 4 had the highest number of QTL exceeding the threshold. In contrast, no QTLs exceeding the threshold of 2.5 were detected in the BLUP and 19DH environment for sub-pop4. However, 99 QTLs exceeding this threshold were identified in the remaining five environments and were distributed across chromosomes 1, 2, 4, and 5, with the highest number of QTLs detected on chromosome 4 (Appendix A). Among the five environments and BLUP, mk388 on chromosome 4 in 21JH exhibited the highest LOD (5.657). Notably, 52 stable QTLs were consistently mapped in the five environments and BLUP in the two subpopulations (Appendix A). The sub-pop3/22YS environment and sub-pop3/BLUP shared seven QTLs. For the pop4 subpopulation, the sub-pop4/21JH and sub-pop4/21YS environments shared one QTL, whereas the sub-pop4/19BS environment shared another QTL.

Across the five environments and BLUPs in the two subpopulations, we identified 677 genes, some of which were repeatedly identified in different subpopulations and some in different environments in the same subpopulation (Table 4). For instance, on chromosome 4, 648 genes were detected in sub-pop3 in the 21JH environment and sub-pop4 in 22YS environment. In sub-pop3 in the 22YS environment, 29 co-located genes were identified on chromosome 4 using BLUP. In some cases, significant QTLs were identified, but no candidate genes were detected. For instance, sub-pop3 in the 19BS environment and sub-pop4 in the 21YS environment shared a common QTL within the interval of 120,350,778–120,390,778 bp on chromosome 5. However, no gene was detected at this position in the B73 reference genome. Nevertheless, when the Mo17 reference genome was used, we successfully detected the gene *Zm00014a012929* within this QTL. This result suggests that when no candidate gene is detected in the B73 reference genome, an alternate reference genome can potentially be used to identify candidate genes associated with significant QTL.

### 2.6. Joint Analysis of GWAS and QTL

By comparing the genes identified via GWAS with those obtained from QTL mapping, we identified 18 genes consistently associated with KRN in both analyses (Table 5). Among these genes, 11 were located on chromosome 1, two on chromosome 4, and the remaining five on chromosome 5. These 18 genes were consistently identified using GWAS and QTL analyses in two or more environments. For instance, in GWAS analysis conducted in the 21JH environment and QTL analysis in the pop3/19DH environment, 11 overlapping genes (*Zm00001d031666*, *Zm00001d031667*, *Zm00001d031668*, *Zm00001d031669*, *Zm00001d031709*, *Zm00001d031713*, *Zm00001d031715*, *Zm00001eb036870*, *Zm00001eb036890*, *Zm00001d031648*, and *Zm00001d031772*) were identified. The phenotypic variation explained (PVE) by these genes ranged from 4.73% to 8.41%. Furthermore, GWAS analysis in the 21JH environment and QTL analysis in the pop3/21JH and pop4/22YS environments identified two genes (*Zm00001d050984* and *Zm00001d050985*) that were consistently associated with KRN, which explained 6.7% of the PVE for KRN. Similarly, GWAS analysis in the 21JH environment and QTL analysis in the pop4/21YS environment revealed four genes (*Zm00001d016000*, *Zm00001d016202*, *Zm00001d016203*, and *Zm00001d016204*) that were consistently linked to KRN, explaining 8.4% to 10.4% of the phenotypic variation for KRN. Interestingly, one molecular marker showed a significant association with KRN across the three environments through GWAS analysis in the 19DH environment, and QTL analysis in the pop3/19BS and pop4/21YS environments. However, no candidate genes were identified when the B73 genome reference was used for analysis. Intriguingly, when the Mo17 reference genome was used, a gene associated with KRN, *Zm00014a012929*, was identified, which explained 8.83% of the KRN variation. This underscores the importance of considering different reference genomes in candidate gene annotation, as it can provide valuable insights into the genes that regulate KRN.

To further validate the 18 co-located candidate genes, we examined their functions and expression levels in the maizeGDB (Table 6). The data in Table 5 showed that *Zm00001d050984* and *Zm00001d050985* were consistently identified in all three environments through either GWAS or QTL analysis. *Zm00001d050985* is expressed in all ear-related tissues and encodes the Ultraviolet-B (UV-B) receptor UVR8, which serves as a light sensor in plants, detecting UV-B wavelengths. UVR8 regulates plant responses to UV-B light and promotes plant growth and development [57,58]. *Zm00001d050984* belongs to the Rab GDI displacement factor A 2 (CHML) gene family [59,60]. It functions as a component of the Rab geranylgeranyltransferase (RGGT) complex, which is responsible for post-translational modifications by adding geranylgeranyl groups to the C-termini of Rab proteins. Although *Zm00001d050984* was not significantly expressed in formation-related tissues, it shared the same SNP as *Zm00001d050985*. Furthermore, approximately 1 Mb downstream of this SNP, there is an fea2 (fasciated ear 2) gene known to cause flattened cobs and irregular KRNs [61,62,63]. Therefore, we considered *Zm00001d050984* and *Zm00001d050985* as potential candidate genes that influence KRN. *Zm00001d016000* belongs to the Myb family of transcription factors. The 1R (R1/2, R3-MYB) subgroup of Myb transcription factors is primarily involved in various physiological processes such as plant morphology, secondary metabolism, circadian rhythm control, and flower fruit development [64]. It also plays a crucial role in diverse physiological processes, including flower organ development and pollen germination [65,66,67,68]. However, based on comprehensive analyses of gene function and expression levels [69,70,71,72], no additional information was found for the other seven genes (Table 6).

### 2.7. Haplotype Analysis

Based on the co-localized candidate gene information from GWAS and QTL analyses and the duplicated pathways obtained from KEGG/GO enrichment, we selected three candidate genes for haplotype analysis. *Zm00001d050984* and *Zm00001d050985*, which were co-located in the three environments, and *Zm00001d016000*, which may be directly related to KRN, were selected for haplotype analysis. *Zm00001d016000* had three haplotypes (hap-1, GGAGG; hap-2, CGAGG; and hap-3, GACAA), and there were no significant differences among these three haplotypes. *Zm00001d050984* had two haplotypes, hap-1 (CC) and hap-2 (TT); however, the difference between the two haplotypes was not significant. *Zm00001d050985* shares the same haplotype pattern as *Zm00001d050884*, and the difference between the two haplotypes was also not significant. Different haplotypes of the same gene may affect gene function, thereby influencing maize KRN. When there are no significant differences among the various haplotypes, they may also co-regulate the gene and make the same contribution to gene expression (Figure 8).

### 2.8. Gene Ontology (GO) and Kyoto Encyclopedia of Genes and Genomes (KEGG) Enrichment Analysis

To precisely identify KRN-associated genes among the large number of candidate genes, we conducted GO and KEGG enrichment analyses to further narrow down the candidate genes identified by GWAS and linkage analysis. GO enrichment analysis provided multiple GO terms, covering three hierarchical levels: cellular components (CC), molecular functions (MF), and biological processes (BP) [81]. The results obtained from GO and KEGG analyses were encouraging. In sub-pop3 for the BLUP and 22YS environments, despite the identification of 29 overlapping genes (Table 4), GO enrichment analysis did not yield significant bioprocesses associated with these genes. Therefore, these 29 genes may not be considered KRN-associated candidate genes. In the case of the sub-pop3 in the 21JH environment and sub-pop4 in the 22YS environment, 648 co-located genes (Table 4) participated in 271 biological processes, primarily related to transport and metabolic processes. These genes were associated with 77 cellular components, including the Golgi apparatus, complexes, and other organelles, and 168 molecular functions involving bioregulation and enzyme activity (Appendix A and Appendix A). In terms of biological processes (BP), the highly correlated category was transmembrane transport, which includes processes such as “nucleotide-sugar transmembrane transport (GO:0015780)” and “pyrimidine nucleotide-sugar transmembrane transport (GO:0090481)”. In cellular components (CC), the highly correlated category was complexes, including the “catalytic complex (GO:1902494)”, “ubiquitin ligase complex (GO:0000151)”, “dynein complex (GO:0030286)”, “respiratory chain complex (GO:0098803)”, and “SAGA-type complex (GO:0070461)”, among others. For molecular function (MF), the highly correlated category included “catalytic activity, specifically acting on a glycoprotein (GO:0140103)”, “nucleotide-sugar transmembrane transporter activity (GO:0005338)”, and “pyrimidine nucleotide-sugar transmembrane transporter activity (GO:0015165)” (Table 7).

After comparing the genes identified from GWAS and QTL analyses with GO annotations, we identified the gene *Zm00001d050984*, which was associated with multiple GO terms. This gene was linked to ten GO terms encompassing both BP and MF. In BP, it participates in “intracellular signal transduction (GO:0035556), signal transduction (GO:0007165), signaling (GO:0023052), small GTPase-mediated signal transduction (GO:0007264), and cell communication (GO:0007154)”, indicating its involvement in various signaling pathways. In MF, it is associated with “nucleoside-triphosphatase regulator activity (GO:0060589), GTPase regulator activity (GO:0030695), GDP-dissociation inhibitor activity (GO:0005092), enzyme regulator activity (GO:0030234), and molecular function regulator (GO:0098772)”, suggesting its role in regulating enzyme activity (Table 8). This *Zm00001d050984* gene appears to be crucial for coordinating multiple signaling pathways and regulating enzyme activity, highlighting its importance in cellular function and response to external stimuli.

After identifying the key cellular components, molecular functions, and biological processes associated with these genes, we conducted KEGG enrichment analysis to reveal their intricate relationships and networks. The results showed that the sub-pop3 for the BLUP and 22YS environments shared genes enriched in eight metabolic pathways, and the most highly correlated categories include “fatty acid elongation (zma00062)” and “ascorbate and aldarate metabolism (zma00053)”. However, the category with the highest number of participating genes was “biosynthesis of cofactors (zma01240)”. Those in sub-pop3 for the 21JH environment and sub-pop4 for the 22YS environment were enriched in 72 metabolic pathways (Appendix A and Appendix A). Some of the highly correlated categories included “butanoate metabolism (zma00650)”, “monoterpenoid biosynthesis (zma00902)”, and “circadian rhythm—plant (zma04712)” (Table 9). The authors suggested that genes associated with these pathways could be considered key candidate genes for further validation.

Through GO and KEGG enrichment analyses, we compared the pathway information with co-located genes identified by GWAS–QTL analyses. In addition to *Zm00001d050984*, which has multiple GO terms, we selected two additional genes, *Zm00001d016202* and *Zm00001d050985. Zm00001d016202* encodes the E3 ubiquitin protein ligase COP1. This gene coincides with the “ubiquitin ligase complex (GO:0000151)” term in the GO analysis of sub-pop3 in the 21JH environment and sub-pop4 in the 22YS environment. *Zm00001d050985* encodes a UV-B wavelength receptor, which might be associated with the “Circadian rhythm—plant (zma04712)” term in the KEGG analysis for the sub-pop3 in the 21JH environment and sub-pop3 in the 22YS environment. It is possible that UVR8 signaling, particularly in response to UV-B radiation, indirectly affects circadian clock regulation in plants.

## 3. Discussion

### 3.1. 18 Co-Located Genes Were Confirmed to Be Associated with KRN

Linkage analysis and GWAS have been widely used to identify candidate genes associated with quantitative traits [82,83,84]. In this study, we developed a multi-parent population comprising 655 RILs by crossing Ye107 from the Reid heterotic group with CML312 and CML444 from the non-Reid group and YML32 and YML46 from the Suwan heterotic group. By combining the results from GWAS and genetic linkage analyses, we identified 18 candidate genes that were co-located in multiple environments (Table 5). The functions of these genes were consistent with those of the GO- and KEGG-enriched pathways. Although previous researchers have identified KRN-associated QTLs on the same chromosomes as in this study [16,85], the physical locations of these KRN-associated QTLs were different. Therefore, we believe that the candidate genes identified in this study are distinct from those reported in previous studies and represent novel candidate genes associated with KRN. Furthermore, our genetic linkage analysis results indicated that some significant QTLS overlapped with the genes that had already been cloned (Appendix A). This finding enhances the reliability of the results of this study. In Appendix A, the QTL on chromosome 5, localized in sub-pop4/21YS, coincided with *qKRN5.04* identified by An et al. [86] who found that the *Zm00001d016075* gene negatively regulated KRN. Although our SNP–QTL co-localization did not pinpoint this gene, the gene we co-localized, *Zm00001d016000*, was approximately four million bases upstream of *Zm00001d016075* and belonged to the same QTL region. Therefore, we hypothesized that they may share similar functions. Given that *Zm00001d016000* is located approximately four million bases upstream of *Zm00001d016075* and falls within the same QTL region, we speculate that they may possess analogous functions, particularly regarding the regulatory effect on KRN.

### 3.2. Key Candidate Genes Linked to KRN Were Identified through GO and KEGG Analysis

In this study, we conducted a comprehensive analysis to identify and validate candidate genes using Gene Ontology (GO), Kyoto Encyclopedia of Genes and Genomes (KEGG), and gene cloning data. These analyses are instrumental in the selection and validation of potential genes associated with the traits of interest (KRN) in maize.

Previous studies have demonstrated the effectiveness of GO and KEGG enrichment analyses in screening candidate genes. For example, Tang et al. [87] used RNA-seq to analyze the transcriptomes of near-isogenic lines and identified 1960 differentially expressed genes (DEGs). Subsequently, the authors employed GO classification and KEGG enrichment analysis to reveal the significantly enriched pathways, shedding light on the genetic basis of KRN in maize. Notably, the *Zm00001d020460* gene within the “DNA binding” pathway emerged as a candidate gene. They further constructed an overexpression vector for *Zm00001d020460* of the YE478 haplotype, transformed the maize inbred line B104, and observed a significant reduction in kernel width in the T1 generation of the positive transgenic lines compared to the wild type.

In our study, we initially narrowed down gene selection by identifying identical GO/KEGG pathways in multiple environments. We then compared the functions of these pathways with those of co-localized genes to screen for candidate genes associated with KRN. Notably, the “circadian rhythm—plant” term was significantly associated with the gene *Zm00001d050985*, identified through GWAS and genetic linkage analysis. This gene encodes the UV-B wavelength receptor. Similarly, the “ubiquitin ligase complex” pathway in GO enrichment analysis corresponded to the gene *Zm00001d016202*, also identified in this study, which encodes the E3 ubiquitin protein ligase COP1. Furthermore, the co-located gene *Zm00001d016000* belongs to the Myb family of transcription factors. Although its exact function is yet to be determined, previous research has suggested that different light wavelengths can regulate maize mesocotyl tissue growth, potentially mediated by circadian rhythmic-related genes, thereby affecting tissue plasticity [88,89]. For example, white, blue, and UV-B light can strongly induce anthocyanin accumulation, and the expression of two key structural genes was high in maize mesocotyls [90]. Furthermore, the overexpression of cryptochrome (CRY) OsCRY1b in rice results in the shortening of the petiole and sheath, and the inhibition of elongation by blue light is related to the interaction between OsCRY1 and E3 ubiquitin protein ligase RFWD2 (COP1) [91]. Additionally, MYB transcription factors play an important role in the response of plants to UV-B light [92]. MYB73 and MYB77 interact with UVR8 to regulate lateral root growth, whereas MYB13 induces secondary flavonoid metabolism and cotyledon expansion under UV-B irradiation. Therefore, we believe that these three genes may interact to jointly regulate the number of rows per ear in maize. In summary, our study harnessed the power of GO and KEGG analyses to identify three candidate genes associated with KRN. This not only streamlined the candidate gene selection process but also offered insights into the molecular mechanisms underlying KRN variations. Collectively, these findings collectively contribute to our understanding of the genetic factors that influence KRN in maize.

### 3.3. Alternative Reference Genome May Help Identify Candidate Gene

In this study, one SNP located between 120,350,778–120,390,778 bp on chromosome 5 was consistently identified in three different environments, including GWAS in the 19DH environment and QTL mapping in sub-pop3 for the 19BS environment, as well as in sub-pop4 for the 21YS environment. Interestingly, when the B73 reference genome was used, no genes were detected in proximity to this SNP. We used another commonly used reference genome, Mo17, and identified the *Zm00014a012929* gene in the same region. Therefore, we believe that genes related to KRN in these lines may not be present in the B73 genome. This phenomenon could be attributed to the introduction of unique gene fragments into the temperate germplasm owing to introgression from the tropical germplasm into the temperate germplasm. This finding underscores the potential of tropical germplasms in improving maize germplasm, as it may carry distinct beneficial genes lacking in temperate germplasms. The lines used in this study were from “Suwan, Reid, and non-Reid heterotic groups”, as proposed by our research group. The presence of novel genes that control KRN in maize from these “three heterotic groups” explains why a candidate gene was identified when Mo17 was used as the reference genome. This outcome further validates the accuracy of the “three heterotic group” model proposed by our research team.

Many GWAS and QTL mapping studies in maize have utilized populations that include Mo17 and B73 or their derived lines. Zhang et al. [93] employed a B73×Mo17 (IBM) Syn10 double haploid (DH) population to successfully identify 100 QTLs and 138 SNPs controlling yield-related traits in maize through a combination of GWAS and QTL mapping. In their study, 52 candidate genes were identified, with one candidate gene, SNPSYN806, which is associated with the number of rows per spike (ERN) and is closely linked to SBP transcription factor 7 (*GRMZM2G098557*) [93]. In another study focusing on the interaction between B73×Mo17 populations, Lin et al. constructed a QTL map for starch content and identified a major QTL, *Qsta9.1*, located on chromosome 9. Subsequent GWAS and co-expression network analysis highlighted *GRMZM2G110929* and *GRMZM5G852704* as potential candidate genes influencing the starch content in maize grains [94].

In contrast, some studies independently analyzed Mo17 and B73 and subsequently compared their disparities. For example, Song et al. investigated the responses of susceptible B73 and resistant Mo17 maize strains to aphids at both metabolite and transcriptome levels. Surprisingly, both strains exhibited time-specific responses to aphid infestation, and even in the absence of herbivory effects, significant differences in gene expression were observed between the two strains [95].

It is worth noting that our study adopted a different approach from the previous ones; we used different reference genomes to identify the candidate genes. The results highlighted that the candidate gene *Zm00014a012929* was exclusively obtained using the Mo17 reference genome and was absent when the B73 reference genome was used. This finding suggests that utilizing different maize reference genomes, especially in cases involving diverse maize heterosis groups, could enhance the chances of identifying pertinent candidate genes. Consequently, this approach holds promise for future research endeavors in maize research.

### 3.4. Key Candidate Genes Associated with KRN

In this study, we identified 277 significant QTLs and 38 significant SNPs that play crucial roles in determining KRN in maize. We further annotated the genes located within a 20 kb range upstream and downstream of these QTLs and SNPs, revealing multiple genes related to KRN. After considering factors such as frequency of occurrence, expression pattern, and functional annotation, and supplemented by GO and KEGG analyses, we identified seven candidate genes associated with KRN. During GWAS analysis, a significant SNP, SNP-177304649, was consistently identified in the 19DH environment. This SNP was associated with two genes, *Zm00001d022420* and *Zm00001d022421*, which were expressed in the KRN-related tissues. Additionally, a *bd1* (branchedsilkless1) gene was found downstream of this SNP. A total of 18 genes were identified through co-localization using GWAS and genetic linkage analysis. Among these, four genes were identified as candidate genes related to KRN. First, *Zm00001d016202*, annotated as the E3 ubiquitin-protein ligase COP1 and associated with the “ubiquitin ligase complex” pathway identified through GO annotation, exhibited a phenotypic variation of 10.64% and was expressed in tissues related to KRN. Next, we identified *Zm00001d050985*, a UV-B receptor associated with the “circadian rhythm—plant” pathway using KEGG analysis. *Zm00001d050984*, which is linked to ten GO terms, appears to be crucial for coordinating multiple signaling pathways and regulating enzyme activity, which may be closely related to the formation of KRN. Downstream of *Zm00001d050985* and *Zm00001d050984*, another gene, *fea2* (fasciated ear 2), was found to be associated with KRN and was expressed in KRN-related tissues. *Zm00001d016000*, a transcription factor of the Myb family, is highly expressed in tissues related to KRN [96]. Finally, for *Zm00014a012929*, although the gene function, expression level, and metabolic pathway remain unclear, it was consistently co-located in most environments in this study and was exclusively identified in the Mo17 reference genome. This gene may not only regulate KRN but also support the validation of the “three heterotic group” model proposed by our research group.

In conclusion, these seven candidate genes are likely to be involved in the regulation of KRN formation in maize. Future research should focus on conducting differential gene expression analyses and validating gene functions to confirm their role in KRN regulation in maize. This study not only provides new genetic markers and genomic resources for exploring the genetic structure and molecular mechanism of maize KRN but also lays a critical foundation for validating and cloning functional genes related to maize kernel row number formation.

## 4. Materials and Methods

### 4.1. Experimental Materials

We used Ye107, a foundational inbred line in China, as the common male parent. Additionally, we selected four parental inbred lines, CML312, CML444, YML46, and YML32, each characterized by significant genetic diversity in terms of KRN, as female parents. The hybridization of these four parental inbred lines with the common male parent Ye107 resulted in the development of four F1 hybrids. Subsequently, we conducted eight successive generations of selfing of the four F1s using the single-seed descent method, leading to the establishment of four distinct recombinant inbred line (RIL) subpopulations. The F8 generation subpopulations resulting from crosses between CML312, CML444, YML46, and YML32 with Ye107 were denoted as sub-pop1, sub-pop2, sub-pop3, and sub-pop4, respectively. Each RIL subpopulation comprised approximately 200 recombinant inbred lines, resulting in 655 RIL lines constituting the multi-parent population. Ye107 has a significant position as a foundation maize parent in China. This line and its subsequent inbred lines were categorized under the Reid group, which is a prominent heterotic group in maize breeding in China (Table 10).

### 4.2. Field Experimental Design and Phenotyping

RILs of the multi-parent population were planted in Dehong (DH) and Baoshan (BS) in Yunnan Province in 2019. The experiments were set up in a randomized complete block design (RCBD) with two replications at each location. Each experimental plot consisted of 4 m long rows with 0.70 m inter-row spacing. Plant-to-plant spacing was 25 cm, with 14 plants per row. Spring trials in 2021 and 2022 were conducted in Yanshan, Yunnan Province, using a Latin square design. Each row had a length of 2.5 m, with a space of 25 cm between plants, and there were 14 plants per row, with two replications. During the winter of 2021, the trial was conducted in Jinghong, Yunnan Province using a random block design. Each row had a length of 2.5 m, with 14 plants per row spaced at 25 cm, with two replications. After the maize cobs matured and dried, five ears were randomly selected from each plot to investigate the KRN.

### 4.3. Phenotypic Data Analysis

Excel 2019 and IBM SPSS Statistics 20 were used to conduct the normal distribution, variance, correlation, and coefficient of variation analyses. The coefficient of variation (CV) was calculated using the following formula: CV = (standard deviation SD/mean) × 100%. The R package ‘lem4’ was used to calculate heritability. Heritability (h^2^) was estimated using the formula proposed by Knapp et al. [97]: h^2^ = σg^2^/(σg^2^ + σge^2^/e + σε^2^/re), where σg^2^ represents the genetic variance, σgl^2^ is the variance of the genotype–environment interaction, σe^2^ is the error variance, n is the number of sites, and y is the number of years.

### 4.4. Genotyping

The parental lines and RILs of the multi-parent population were genotyped via the genotyping-by-sequencing (GBS) method. DNA was extracted from 655 RILs of the four subpopulations, along with their respective parents, using the method described by Elshire et al. [98]. After the DNA library of 300 bp size was constructed and sequenced on the Illumina NovaSeq 6000 platform (Illumina Inc., San Diego, CA, USA) with a read length of 2 × 150 bp, the raw sequencing reads were processed to remove reads with adapters and low-quality reads, resulting in high-quality filtered reads. Single nucleotide polymorphisms (SNPs) and insertions and deletions (InDels) were detected using GATK software v4.1.4.0. Subsequent association analysis was performed using 590,816 high-quality SNPs.

### 4.5. Structure Analysis

Phylogenetic tree analysis was conducted using Tassel v5.0 software, utilizing 590,816 high-quality SNPs to assess the genetic relationships among the 655 RILs. Principal component analysis was performed using R 4.3.2, and the results were visualized using the scatterplot3d software package.

### 4.6. Genome-Wide Association Analysis

Genome-wide association analysis was conducted using the GEMMA software (open model: https://github.com/genetics-statistics/GEMMA/releases, accessed on 1 June 2023) [99], and 590,816 high-quality SNPs were used in the analysis. The analysis employed a mixed linear model (MLM) for GWAS for both the average and BLUP values for all environments. Significance was determined at the *p* < 0.0001 level using the formula −log10 (1/Total number of SNPs) to assess the association between SNP markers and panicle number. The distribution of markers is shown in the Manhattan plot, and the Q-Q plot was used to evaluate the accuracy of the association analysis results.

### 4.7. Haplotype Analysis

Haploview v4.2 software was used to analyze genes that were consistently detected in multiple environments whose functions were related to KRN.

### 4.8. Candidate Gene Screening

SNPs obtained from GWAS results were annotated using the ANNOVAR software (v2013-05-20). Based on the annotated gene numbers, the SNPs were searched and compared in maizeGDB and NCBI (http://www.ncbi.nlm.nih.gov/ (accessed on 15 September 2023)) databases, leading to the identification and functional annotation of candidate genes.

### 4.9. Construction of Genetic Linkage Map

The JoinMap4.0 software was used for linkage map construction. The LOD threshold was determined using 1000 random permutation tests, with a significance level of *p* ≤ 0.05 [100]. QTLs were considered significant if the LOD threshold was ≥2.5. Linkage groups were defined based on the criteria of an LOD critical value ≥ 2.5, and a genetic linkage map was constructed using eligible SNP markers. SNP markers within each linkage group were sorted using the maximum likelihood method and unlinked markers were excluded. The genetic distance (cM) between markers was computed using the Kosambi mapping function. Polymorphic SNPs with distinct alleles in both parents were used to construct genetic maps of sub-pop3 and sub-pop4. Markers showing segregation distortion were filtered out using the Chi-square test (*p* < 0.001), and markers with an integrity <95% were also excluded.

### 4.10. QTL Mapping Analysis

QTL mapping was conducted using the composite interval mapping (CIM) method in Windows QTL Cartographer 2.0. Phenotypic data for KRN were integrated to the high-density genetic linkage map, and a step distance of 0.21 cM was used to identify parental polymorphic QTLs. The threshold was established using 1000 random permutations at a 95% confidence interval (CI). The LOD thresholds associated with flanking markers were set to 2.5 to identify QTLs controlling KRN.

### 4.11. GO and KEGG Enrichment Analysis

Based on the results of linkage disequilibrium decay distance analysis, genes located within 20kb upstream and downstream of the significant SNPs were functionally annotated using GO and KEGG analyses. The annotation process helped further validate and identify candidate genes associated with the target traits obtained through GWAS and QTL analyses. The *p*-values for GO and KEGG analyses were calculated using the following formula:$$P=1-\sum_{i=0}^{m-1}\frac{\left(\begin{array}{c} M\\ i \end{array}\right)\left(\begin{array}{c} N-M\\ n-i \end{array}\right)}{\left(\begin{array}{c} N\\ n \end{array}\right)}$$

## Figures and Tables

**Figure 1 ijms-25-03377-f001:**
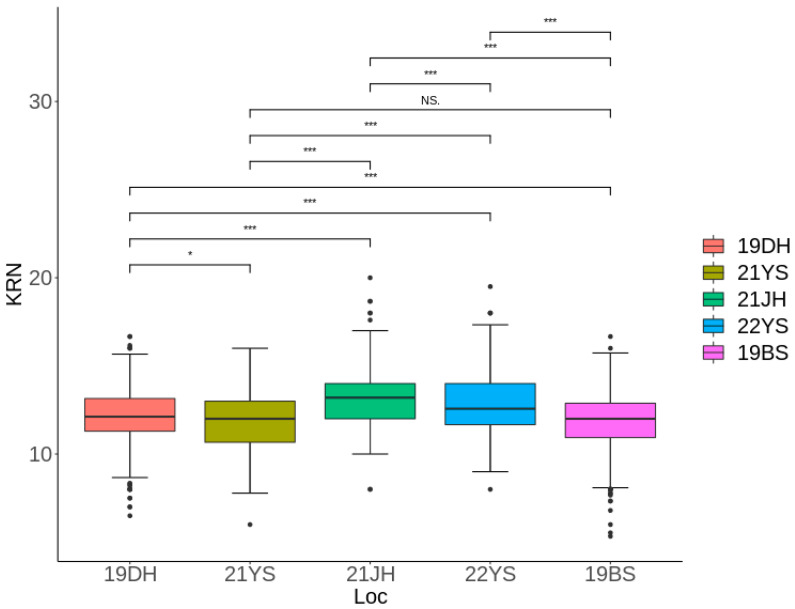
KRN variations in the multi-parent population in the five environments. 19BS for Baoshan in 2019; 19DH for Dehong in 2019; 21JH for Jinghong in 2021; 21YS for Yanshan in 2021; 22YS for Yanshan in 2022. In Figure 1, in the boxplot, the horizontal line in the middle represents the median, the upper and lower lines of the box refer to 75th and 25th quartiles, respectively, and the whiskers for each box indicate the minimum and maximum values within the box plot range (whiskers). Significance levels are denoted as follows: “***” ≤ 0.0001, “*” ≤ 0.01, “NS” ≥ 0.05.

**Figure 2 ijms-25-03377-f002:**
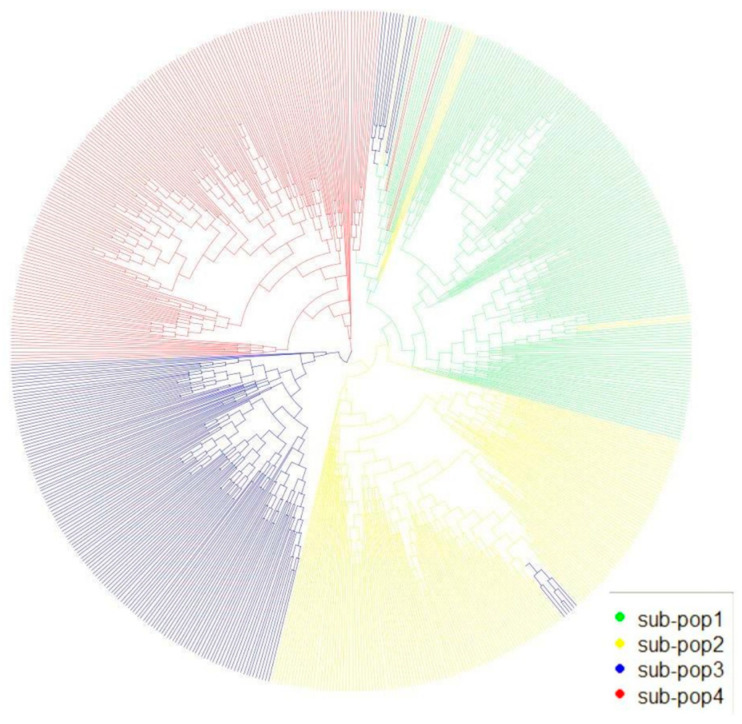
UPGMA analysis of 655 RILs. The color indicates different sub-populations: green: YML32 × Ye107 (sub-pop1); yellow: YML46 × Ye107 (sub-pop2); blue: CML312 × Ye107 (sub-pop3); red: CML444 × Ye107 (sub-pop4).

**Figure 3 ijms-25-03377-f003:**
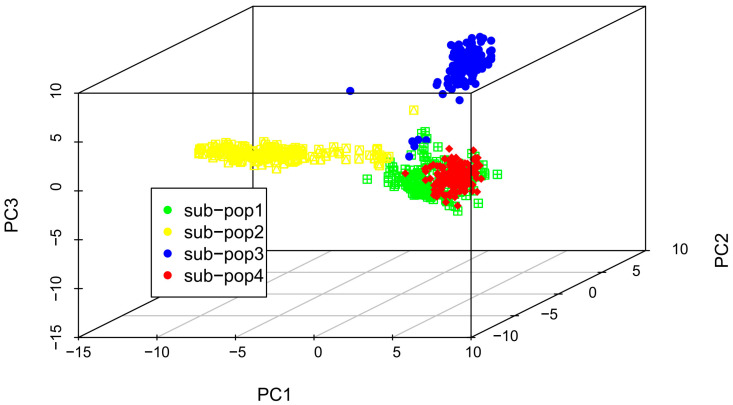
PCA of 655 RILs. The sub-populations are identified as follows: sub-pop1: YML32 × Ye107; sub-pop2: YML46 × Ye107; sub-pop3: CML312 × Ye107; sub-pop4: CML444 × Ye107.

**Figure 4 ijms-25-03377-f004:**
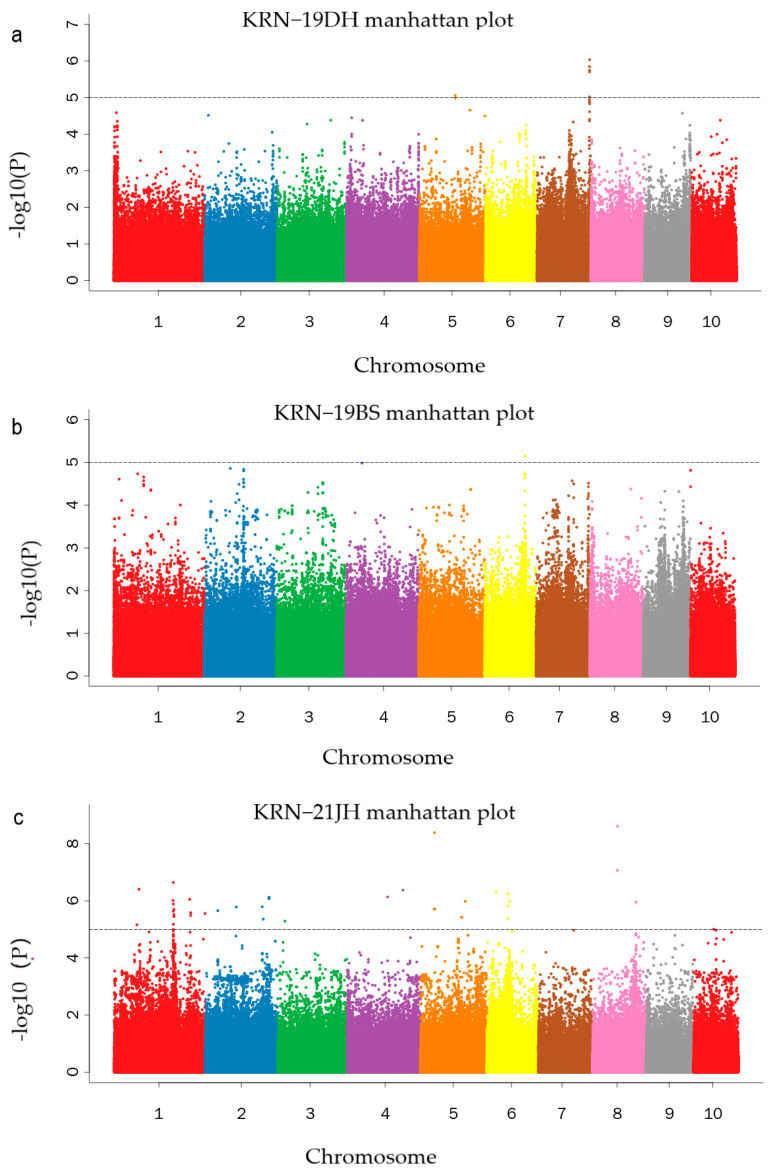
Manhattan plots: (**a**) Dehong 2019 (19DH), (**b**) Baoshan 2019 (19BS), (**c**) Jinghong 2021 (21JH), (**d**) Yanshan 2021 (21YS), and (**e**) Yanshan 2022 (22YS).

**Figure 5 ijms-25-03377-f005:**
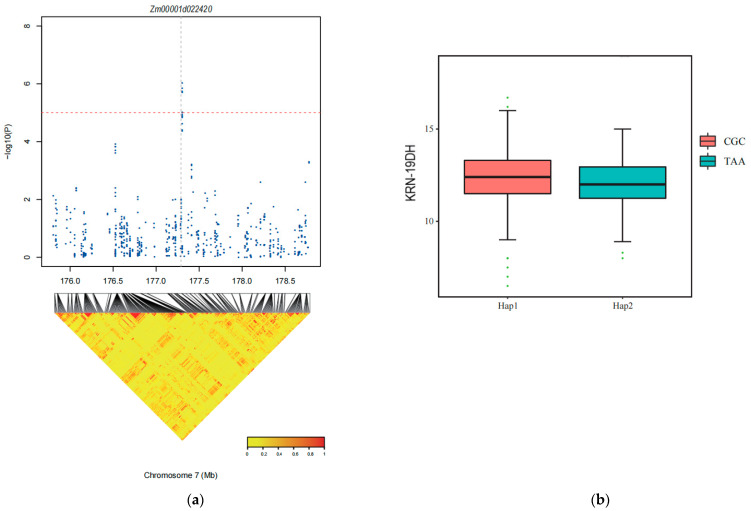
Haplotype analysis of *Zm00001d022420* and *Zm00001d022421*. (**a**) The position of the SNPs associated with *Zm00001d022420* identified by GWAS. (**b**) The differences in KRN between the two haplotypes (GG, AA) in the 19DH environments. (**c**) Haplotype analysis of *Zm00001d022421* and the SNP position on chromosome 7. (**d**) Differences in KRN between the two haplotypes (CGC, TAA) in the 19 DH environment. “***”: *p* ≤ 0.001. (**e**) The locations of candidate genes associated with KRN.

**Figure 6 ijms-25-03377-f006:**
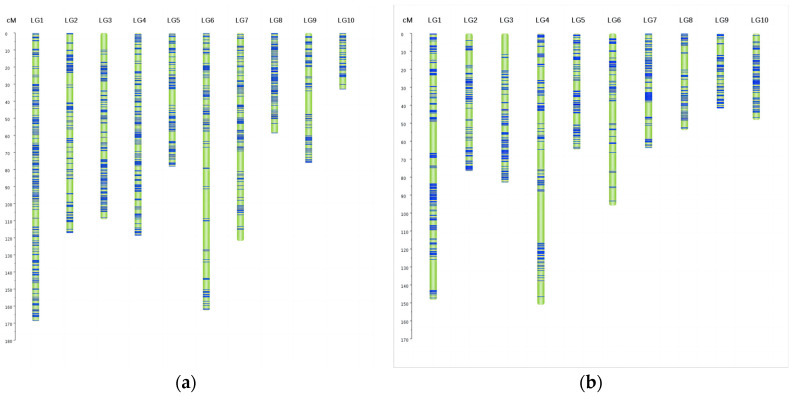
Genetic linkage maps of sub-pop3 (**a**) and sub-pop4 (**b**). Chromosomes 1 to 10 are labeled as LG1 to LG10, respectively. The blue lines represent the bins, indicating marker segments spaced every 15 markers without linkage, while the green spaces between bins represent the distances between adjacent bins.

**Figure 7 ijms-25-03377-f007:**
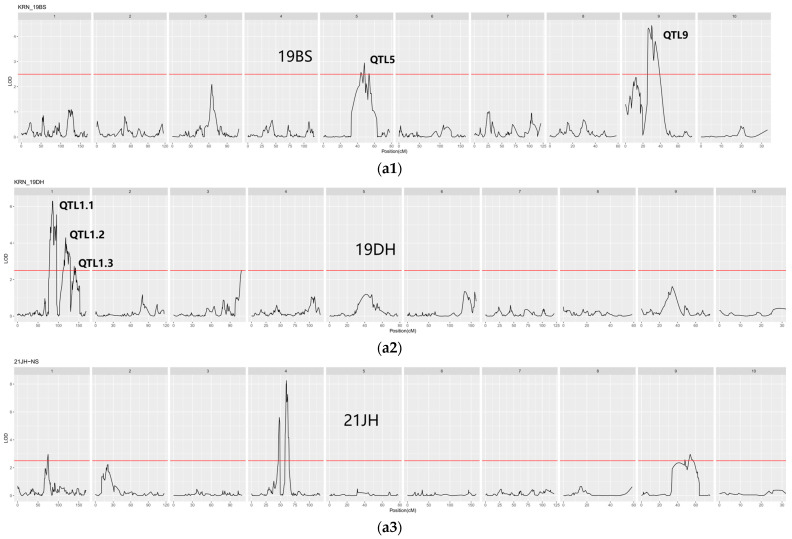
QTL analysis of two subpopulations in five different environments and BLUP. (**a1–a6**) QTL mapping in sub-pop3 in five different environments and BLUP. (**b1–b6**) QTL mapping in sub-pop4 in five different environments and BLUP. The red line represents the LOD threshold in subfigures (**a**,**b**) The best linear unbiased prediction over all environments (BLUP), merged value of Yanshan in 2022 (22YS), Yanshan in 2021 (21YS), Dehong in 2019 (19DH), Baoshan in 2019 (19BS), and Jinghong in 2021 (21JH).

**Figure 8 ijms-25-03377-f008:**
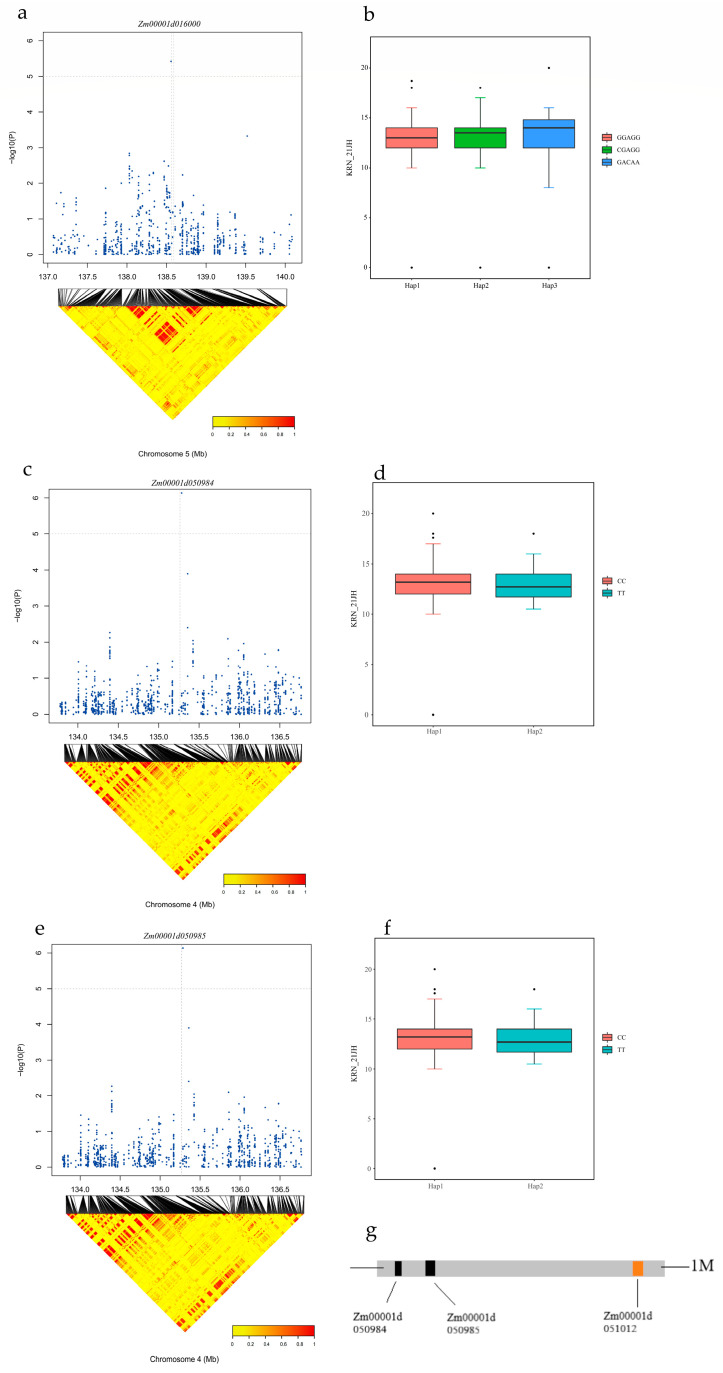
Haplotype analysis of Zm00001d050984, Zm00001d050985, and Zm00001d016000. (**a**) Positions of the SNPs in the candidate gene associated with Zm00001d016000 on chromosome 5; (**b**) Differential effects of three haplotypes (GGAGG, CGAGG, and GACAA) of Zm00001d016000 on KRN; (**c**) Positions of the SNPs in the candidate genes associated with *Zm00001d050984* on chromosome 4; (**d**) Differential effects of CC/TT haplotypes of *Zm00001d050984* on KRN; (**e**) Positions of the SNPs in the candidate genes associated with *Zm00001d050985* on chromosome 4; (**f**) Differential effects of CC/TT haplotypes of *Zm00001d050985* on KRN; (**g**) Candidate gene positions associated with KRN.

**Table 1 ijms-25-03377-t001:** Phenotypic variation in KRN across five environments in the multi-parent population ^†^.

Environment	Min	Max	AVE	SD	CV(%)	H^2^ (%)
19DH	6.5	16.67	12.1647	1.46912	12.07691106	76.8
19BS	5.33	16.67	11.8247	1.54526	13.06806938	
21YS	6	16	11.9313	1.83939	15.41650951	
21JH	8	20	13.2378	1.59104	12.01891552	
22YS	9	19.5	12.7630	1.70710	13.37538196	

^†^, CV: coefficient of variation; H2: broad-sense heritability; SD: standard deviation.

**Table 2 ijms-25-03377-t002:** Genes associated with SNPs explaining >10% KRN variation and their expression in tissues relevant to KRN ^†^ [40,41,42,43,44,45,46,47,48,49,50,51].

Gene ID	Env	SNP	Chr	PVE (%)	Function	Expression (FPKM)			
						Meristem_16–19	Ear_Primordium (2–4 mm)	Ear_Primordium (6–8 mm)	Female_Spikelet
*Zm00001d007924*	21YS	242858336	2	17.74	Cytochrome P450 41	NA	NA	NA	NA
*Zm00001d007925*					Cyclic nucleotide-gated ion channel 18	NA	NA	NA	NA
*Zm00001d025694*	22YS	126528998	10	14.88	Surfeit locus protein 2 [52]	58.5	73.9	63	35.7
*Zm00001d025695*					Tetratricopeptide-like helical domain superfamily	1.0	1.2	1.4	2.2
*Zm00001d025696*					Cellulase (Glycosyl hydrolase family 5) Protein	4.1	5.0	5.1	6.0
*Zm00001d029702*	21JH	83163645	1	21.01	glutathione S-transferase	NA	NA	NA	NA
*Zm00001d006733*	21JH	215468689	2	14.31	Transcription factor bHLH85	NA	NA	NA	NA
*Zm00001d006734*					Zinc finger protein AZF2	Not reported			
*Zm00001d006735*					Zinc finger protein AZF1				
*Zm00001d014435*	21JH	46725146	5	10.57	D-xylose-proton symporter-like 3	58.5	49.8	46	29.3
*Zm00001d016202*	21JH	150393177	5	10.64	E3 ubiquitin-protein ligase COP1	0.1	0.2	0.2	NA
*Zm00001d016203*					Actin-related protein 2/3 complex subunit 3	20.3	11.9	11.6	11.1
*Zm00001d016204*					Putative ubiquitin-like-specific protease 1B	Not reported			
*Zm00001d024390*	21JH	68447065	10	11.15	Sec14p-like phosphatidylinositol transfer family protein	3.6	8.8	9.1	NA

^†^, NA indicates that the gene is not expressed in the corresponding tissue. Env, environment; SNP, single nucleotide polymorphisms; Chr, chromosome; PVE, percentage of phenotypic variation explained.

**Table 3 ijms-25-03377-t003:** Genes corresponding to SNP-177304649 and their expression levels in KRN-related tissues ^†^.

Gene	Loc	SNP	Chr	PVE (%)	Function	Expression (FPKM)			
						Meristem_16–19_Day	Ear_Primordium (2–4 mm)	Ear_Primordium (6–8 mm)	Female_Spikelet
*Zm00001d022420*	19DH	177304649	7	6.79	ricin B-like lectins	NA	NA	NA	61.5
*Zm00001d022421*						122.7	156	124.5	464.8

^†^, NA indicates that the gene is not expressed in the corresponding tissue. Meristem_16–19_Day: Meristem during days 16–19.

**Table 4 ijms-25-03377-t004:** Number of genes repeatedly identified across different environments ^†^.

Loc	Chr	Start	End	Number of Repeated Genes
sub-pop3/21JH	4	75,627,569	151,339,536	648
sub-pop4/22YS	4	46,403,895	172,368,609	
sub-pop3/22YS	4	8,043,739	18,953,659	29
sub-pop3/BLUP	4	8,043,739	10,683,376	
sub-pop4/21JH	4	31,671,021	31,671,020	0
sub-Pop4/21YS	4	31,409,140	31,409,140	
Total				677

^†^, Loc: location. Chr: chromosomes. Start and end: Starting and ending position of the QTL.

**Table 5 ijms-25-03377-t005:** Co-localized genes identified through GWAS and QTL mapping ^†^.

Gene ID	PVE (%)	GWAS-Loc	Start	End	SNP	Chr	QTL-Loc	
*Zm00001d031666*	7.02	21JH	198,109,347	198,149,347	3	1	312/19DH	
*Zm00001d031667*								
*Zm00001d031668*								
*Zm00001d031669*								
*Zm00001d031709*	6.14	21JH	199,086,836	199,126,836	4	1	312/19DH	
*Zm00001d031713*	8.41	21JH	199,202,030	199,242,030	5	1	312/19DH	
*Zm00001d031715*								
*Zm00001eb036870*	6.48	21JH	199,294,176	199,334,176	6	1	312/19DH	
*Zm00001eb036890*	4.73	21JH	199,315,283	199,355,283	7	1	312/19DH	
*Zm00001d031648*								
*Zm00001d031772*	5.57	21JH	201,335,723	201,375,824	8	1	312/19DH	
*Zm00001d050984*	6.70	21JH	135,263,055	135,303,055	1	4	312/21JH	444/22YS
*Zm00001d050985*								
*Zm00001d016000*	8.40	21JH	138,534,600	138,574,630	2	5	444/21YS	
*Zm00001d016202*	10.64	21JH	150,373,177	150,413,177	1	5	444/21YS	
*Zm00001d016203*								
*Zm00001d016204*								
*Zm00014a012929*	8.83	19DH	120,35,0778	120,390,778	1	5	312/19BS	444/21YS

^†^, PVE: phenotypic variance explained. GWAS-Loc: environment in GWAS. Start and end: Starting and ending position of the QTL. SNP: Number of Single Nucleotide Polymorphism in the interval from start to end. Chr: chromosome. QTL-Loc: Environment in QTL.

**Table 6 ijms-25-03377-t006:** Functional annotation and expression levels of co-located genes ^†^ [35,40,42,67,68,71,72,73,74,75,76,77,78,79,80].

Gene	Chr	Function	Expression (FPKM)			
			Meristem_16–19	Ear_Primordium (2–4 mm)	Ear_Primordium (6–8 mm)	Female_Spikelet
*Zm00001d031666*	1	Probable aldo-keto reductase 2	2.1	4.3	4.5	6.2
*Zm00001d031667*	1	Discolored-paralog2	13.3	12.3	15.2	7.8
*Zm00001d031668*	1	Ubiquitin thioesterase otubain-like	8.9	5.2	6.4	7.3
*Zm00001d031669*	1	O-fucosyltransferase family protein	12.3	8.9	8.7	8.8
*Zm00001d031709*	1	Replication protein A 70 kDa DNA-binding subunit B	NA	NA	NA	NA
*Zm00001d031713*	1	CBL-interacting protein kinase 19	NA	NA	NA	NA
*Zm00001d031715*	1	CBL-interacting serine/threonine-protein kinase 9	NA	1.3	1.8	NA
*Zm00001eb036870*	1	Unknown	Unreported			
*Zm00001eb036890*	1	Protein S-acyltransferase4				
*Zm00001d031648*	1	Protein S-acyltransferase4	47.5	26.6	26.2	64.8
*Zm00001d031772*	1	Protein NtpR	NA	NA	NA	NA
*Zm00001d050984*	4	Rab-proteins geranylgeranyltransferase component A	NA	0.2	0.2	NA
*Zm00001d050985*	4	Ultraviolet-B receptor UVR8	8.7	18.8	18.1	15.7
*Zm00001d016000*	5	Myb-related protein 3R-1	52.3	55.5	59.3	2.6
*Zm00001d016202*	5	E3 ubiquitin-protein ligase COP1	NA	NA	NA	NA
*Zm00001d016203*	5	Actin-related protein 2/3 complex subunit 3	20.3	11.9	11.6	11.1
*Zm00001d016204*	5	Putative ubiquitin-like-specific protease 1B	NA	NA	NA	NA
*Zm00014a012929 (mo17)*	5	Unknown	Unreported			

^†^ “NA” in the candidate genes column indicates currently unidentified genes.

**Table 7 ijms-25-03377-t007:** The highly correlated Gene Ontology (GO) terms ^†^.

LOC	Category	GO ID	Description	Gene Ratio	*p* Value
Sub-pop3/21JH and sub-pop4/22YS	BP	GO:0015780	Nucleotide-sugar transmembrane transport	2/170	0.011210655
	BP	GO:0090481	Pyrimidine nucleotide-sugar transmembrane transport	2/170	0.011210655
	BP	GO:0006914	Autophagy	2/170	0.013554188
	BP	GO:0061919	Process utilizing autophagic mechanism	2/170	0.013554188
	BP	GO:0006325	Chromatin organization	4/170	0.020186478
	BP	GO:1901264	Carbohydrate derivative transport	2/170	0.021710384
	BP	GO:0015698	Inorganic anion transport	3/170	0.024541102
	BP	GO:0006821	Chloride transport	2/170	0.034964547
	BP	GO:0006497	Protein lipidation	2/170	0.054807613
	BP	GO:0006505	GPI anchor metabolic process	2/170	0.054807613
	CC	GO:0000139	Golgi membrane	2/59	0.02194791
	CC	GO:0098791	Golgi subcompartment	2/59	0.025312478
	CC	GO:0031984	Organelle subcompartment	3/59	0.0600554
	CC	GO:1902494	Catalytic complex	7/59	0.103515469
	CC	GO:0005794	Golgi apparatus	2/59	0.155306753
	CC	GO:0044431	Golgi apparatus part	2/59	0.155306753
	CC	GO:0000151	Ubiquitin ligase complex	1/59	0.166579053
	CC	GO:0030286	Dynein complex	1/59	0.166579053
	CC	GO:0098803	Respiratory chain complex	1/59	0.166579053
	CC	GO:0070461	SAGA-type complex	1/59	0.181653191
	MF	GO:0140103	Catalytic activity, acting on a glycoprotein	3/239	0.001729669
	MF	GO:0005338	Nucleotide-sugar transmembrane transporter activity	2/239	0.009752157
	MF	GO:0015165	Pyrimidine nucleotide-sugar transmembrane transporter activity	2/239	0.009752157
	MF	GO:0033926	Glycopeptide alpha-N-acetylgalactosaminidase activity	2/239	0.011799431
	MF	GO:0015103	Inorganic anion transmembrane transporter activity	3/239	0.016322483
	MF	GO:0008509	Anion transmembrane transporter activity	3/239	0.023303348
	MF	GO:0005244	Voltage-gated ion channel activity	2/239	0.030592768
	MF	GO:0005247	Voltage-gated chloride channel activity	2/239	0.030592768
	MF	GO:0005253	Anion channel activity	2/239	0.030592768
	MF	GO:0005254	Chloride channel activity	2/239	0.030592768

^†^, Category: categories of GO annotations, including molecular function (MF), cellular component (CC), and biological process (BP). Gene Ratio: a fraction where the numerator is the number of genes enriched in this GO term, and the denominator is the total number of genes used for enrichment analysis. LOC: location. Sub-pop3/21JH: pop3 subpopulation for the 21JH environment. Sub-pop4/22YS: pop4 subpopulation for the 22YS environment.

**Table 8 ijms-25-03377-t008:** The Gene Ontology (GO) terms associated with *Zm00001d050984* ^†^.

Gene	GWAS-Loc	QTL Loc	Description	GO ID	Category
*Zm00001d050984*	21JH	Sub-pop3/21JH and sub-pop4/22YS	intracellular signal transduction	GO:0035556	BP
			signal transduction	GO:0007165	BP
			signaling	GO:0023052	BP
			small GTPase mediated signal transduction	GO:0007264	BP
			cell communication	GO:0007154	BP
			nucleoside-triphosphatase regulator activity	GO:0060589	MF
			GTPase regulator activity	GO:0030695	MF
			GDP-dissociation inhibitor activity	GO:0005092	MF
			enzyme regulator activity	GO:0030234	MF
			molecular function regulator	GO:0098772	MF

^†^, Category: categories of GO Annotations, including molecular function (MF) and biological process (BP). 21JH: Jinghong in 2021. Sub-pop3/21JH: the pop3 subpopulation for the 21JH environment. Sub-pop4/22YS: the pop4 subpopulation for the 22YS environment.

**Table 9 ijms-25-03377-t009:** The Kyoto Encyclopedia of Genes and Genomes (KEGG) associated with *Zm00001d050984* ^†^.

Loc	KEGGID	Description	Bg Ratio	*p* Value
Sub-Pop3/21JH and sub-pop4/22YS	zma00650	Butanoate metabolism	30/5493	0.010066583
	zma00902	Monoterpenoid biosynthesis	12/5493	0.013495559
	zma00770	Pantothenate and CoA biosynthesis	40/5493	0.021995592
	zma00020	Citrate cycle (TCA cycle)	72/5493	0.022923456
	zma00290	Valine, leucine and isoleucine biosynthesis	22/5493	0.042859117
	zma00563	Glycosylphosphatidylinositol (GPI)-anchor biosynthesis	24/5493	0.050229291
	zma04712	Circadian rhythm—plant	58/5493	0.056694817
	zma01210	2-Oxocarboxylic acid metabolism	61/5493	0.064061405
	zma01200	Carbon metabolism	290/5493	0.070207933
	zma01040	Biosynthesis of unsaturated fatty acids	29/5493	0.070421369
Sub-pop3/22YS and sub-pop4/BLUP	zma00062	Fatty acid elongation	40/5493	0.028819143
	zma00053	Ascorbate and aldarate metabolism	60/5493	0.042992849
	zma00040	Pentose and glucuronate interconversions	77/5493	0.054918027
	zma01250	Biosynthesis of nucleotide sugars	105/5493	0.074316114
	zma00480	Glutathione metabolism	111/5493	0.07843369
	zma00520	Amino sugar and nucleotide sugar metabolism	160/5493	0.111548551
	zma04626	Plant-pathogen interaction	228/5493	0.156015338
	zma01240	Biosynthesis of cofactors	265/5493	0.179497585

^†^, Sub-pop3/21JH: the pop3 subpopulation for the 21JH environment; sub-pop4/22YS: the pop4 subpopulation for the 22YS environment; sub-pop3/22YS and sub-pop3/BLUP: the pop3 subpopulation for the BLUP and 22YS. BgRatio: The denominator is the total number of genes in the genes encoding proteins of the species wherever they carry GO annotations (estimated to be the total GO gene pool), and the numerator is the number of genes within the total number that carry GO functional annotations on the current entry.

**Table 10 ijms-25-03377-t010:** Parental lines used to develop multi-parent population.

Parental Lines	Pedigree	Heterotic Group	Ecotype
Ye107	From US hybrid DeKalb XL8	Reid	Temperate
CML312	S89500-F2-2-2-1-1-B	Non-Reid	Tropical
CML444	P43-C9-1-1-1-1-1	Non-Reid	Tropical
YML46	Selected from Suwan1	Suwan	Tropical
YML32	Selected from Suwan1	Suwan	Tropical

## Data Availability

The data presented in this study are available on request from the corresponding author.

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
