# Peer review of "Genome-Wide Association Studies on the Kernel Row Number in a Multi-Parent Maize Population"

_ijms, 2024, doi:10.3390/ijms25063377_

Round 1

Reviewer 1 Report

Comments and Suggestions for Authors

The manuscript presents the results of an exploratory search for new loci and genes that could be associated with Kernel Row Number (KRN) in the maize cob, a critical trait that directly affects yield. The authors have done extensive work to obtain and characterize the multi-parent population. Phenotyping for the target trait, accompanied by genotyping and expression analysis of candidate genes, allowed the authors to identify significant KRN-associated QTLs and SNPs. Including, 5 SNPs and 7 novel candidate genes were identified that are highly likely to be associated with KRN.

The work is presented clearly in the manuscript and can be accepted for publication in the IJMS. The results obtained are properly analyzed and discussed fully. The discussion can be supplemented with data from two more recent papers published in 2022, one of which describes a gene that has a negative effect on CRN, and the other a positive one (the latest data is on qKRN5.04). : 10.1111/nph.17882 and 10.1007/s00122-022-04089-w.

Reviewer 2 Report

Comments and Suggestions for Authors

The article is very interesting and focuses on a fundamental issue.

The major issues which need to be addressed are:

Q1: How much enhancement in yield is expected through these mechanisms?

Q2: Have these genes been reported in the genomes of other crops?

Q3: Is this negatively co-related to disease resistance or other abiotic stress?

Q4: Are there any unique nucleotide signatures in these genes?
